# Integrated Pest Management Strategies to Control Varroa Mites and Their Effect on Viral Loads in Honey Bee Colonies

**DOI:** 10.3390/insects15020115

**Published:** 2024-02-05

**Authors:** Jernej Bubnič, Janez Prešern, Marco Pietropaoli, Antonella Cersini, Ajda Moškrič, Giovanni Formato, Veronica Manara, Maja Ivana Smodiš Škerl

**Affiliations:** 1Agricultural Institute of Slovenia, Hacquetova Ulica 17, 1000 Ljubljana, Slovenia; jernej.bubnic@kis.si (J.B.); janez.presern@kis.si (J.P.); ajda.moskric@kis.si (A.M.); maja.smodis.skerl@kis.si (M.I.S.Š.); 2Istituto Zooprofilattico Sperimentale del Lazio e della Toscana “M. Aleandri”, Via Appia Nuova 1411, 00178 Rome, Italy; antonella.cersini@izslt.it (A.C.); giovanni.formato@izslt.it (G.F.); veronica.manara-esterno@izslt.it (V.M.)

**Keywords:** queen caging, trapping comb, *Varroa destructor*, Deformed Wing Virus, Acute Bee Paralysis Virus

## Abstract

**Simple Summary:**

The aim of the study was to evaluate the viral load in honey bee colonies after adopting two brood interruption techniques that are used to control varroa mite. We evaluated the efficacy of two integrated pest management (IPM) strategies, “Queen Caging” (QC) and “Trapping Comb” (TC) procedures, in conjunction with an oxalic acid treatment, to control varroa infestations and consequently lower the viral loads of Deformed Wing Virus (DWV) and Acute Bee Paralysis Virus (ABPV). Two distinct apiaries in Slovenia and Italy, each with a different climate, served as the research sites. In the experiment, the adult bee viral load, mite fall, colony strength, and acaricide efficiency were assessed. The study indicated that the TC approach might be more successful in lowering viral loads. Our results also showed that the acaricidal efficacy of the applied IPM protocols is high. Our study is the first attempt to assess viral infections in honey bees after IPM adoption. The results show the potential advantages of using targeted varroa treatments in combination with brood interruption strategies to manage honey bee viruses vectored by varroa mite.

**Abstract:**

Honey bee viruses in combination with varroa mite are very damaging for honey bee colonies worldwide. There are no effective methods to control the viral load in honey bee colonies except regular and effective control of mites. Integrated Pest Management strategies are required to effectively control mites with veterinary medicines based on organic compounds. We evaluated the effect of two brood interruption techniques, queen caging (QC) and trapping comb (TC), followed by an oxalic acid treatment, on the mite fall, colony strength, and viral load of Deformed Wing Virus (DWV) and Acute Bee Paralysis Virus (ABPV). In this paper, we report the data obtained in two experimental sites, in Slovenia and Italy, in terms of the varroacide efficacy, colony strength, and viral load. The number of adult bees after the adoption of the two techniques showed similar decreasing trends in both locations. The viral load of Acute Bee Paralysis Virus did not show any significant reduction after 25 days, reported as the number of Real-Time PCR cycles needed to detect the virus. The viral load of DWV also did not show a significant reduction after 25 days. The acaricidal efficacy of the applied protocols was high in both experimental groups and in both apiaries. Both the queen caging and trapping comb techniques, followed by an oxalic acid treatment, can be considered effective varroa treatment strategies, but further studies should be carried out to evaluate the long-term effects on viral loads to plan the Integrated Pest Management strategy with the right timing before wintering.

## 1. Introduction

*Varroa destructor* [1] is undoubtedly the most important and damaging pathogen among honey bee (*Apis mellifera*) pests [2]. Mites feed on the fat bodies [3] of bee larvae during the reproductive phase and adults during the dispersal phase, causing severe damage to individual bees and at colony level [4]. Besides the damage caused by feeding, varroa is also a vector of other pathogens like bacteria, fungi, parasitoids, microsporidia, and viruses [5]. A strong relationship between varroa mite and honey bees’ viruses has been shown for some Deformed Wing Virus (DWV) variants [6,7], Acute Bee Paralysis Virus (ABPV), Sacbrood Virus (SBV), Kashmir Bee Virus (KBV), and Israeli Acute Paralysis Virus (IAPV) [8]. Before varroa spread through the population of Western honey bees, viruses were mainly present as covert infections, not causing any clinical signs of disease [9]. Today, clinical signs of viral diseases are a common sight in honey bee colonies, and the overall prevalence of viruses has increased [10,11,12].

DWV is a single-stranded RNA virus from the picorna-like family Iflaviridae [13,14]. Genetically, DWV is very variable. Today, it is classified in many variants, of which DWV-A and DWV-B are the main ones [15]. Recently, it has been demonstrated that some variants can replicate exclusively in *V. destructor* and not in *A. mellifera*, showing how their presence in honey bees is related to the varroa’s feeding behavior [16].

The DWV virus can be transmitted horizontally (e.g., through feces and contaminated feed), vertically (through infected eggs from the infected queen or drone sperm), and by vectors (commonly varroa mite but also *Tropilaelaps* spp. and Small Hive Beetle) [17,18].

There are many unknown peculiarities in the relationship between varroa and DWV, but it is known that, in the pathogenesis of DWV symptoms, the virus must replicate in mites prior to viral transmission to the bee larvae, up to threshold levels (more than 1010 viral genome equivalents per mite) [19]. Several studies have confirmed the positive correlation between the number of mites per pupae, the amount of DWV in individual pupae, and the incidence of clinical symptoms [20].

ABPV was first reported in honey bees in 1963, not causing any clinical signs [21]. Since then, it has spread all over the world [13]. However, it is not highly prevalent in apiaries, and it is influenced by seasonality [22]. The ABPV virus may be spread horizontally by feces [21] and salivary gland secretion [23] or vertically via sperm [24]. After the arrival of varroa mites in Russia and Germany, the titers of ABPV in collapsing colonies increased [10]. It was also shown in vitro that varroa mites could transmit ABPV [10,11]. However, it is still not clear whether ABPV replicates in mites or not [22]. Bee lice, *Braula schmitzi,* can also act as a vector of ABPV [25].

Despite many attempts, there are still not many specific and effective treatments for the viral diseases of honey bees. A promising approach is RNA interference (RNAi) technology, so far used to manage Israeli Acute Paralysis Virus (IAPV) [26,27] and Chinese Sacbrood Virus (CSBV) [28]. However, all the attempts to apply RNAi technology to practice were for experimental purposes. The most widely used method for controlling viral diseases is the destruction of severely damaged colonies or in case of less severe subclinical infections, by effectively controlling the varroa mite population [29,30].

Integrated Pest Management (IPM) is a systemic approach to pest control. It is the process of a combination of different strategies and measures to control pest populations, keeping pesticides and other chemicals to levels that are economically justified, reducing or minimizing risks to human health and the environment [31].

“Queen caging” (QC) and “trapping comb” (TC) are beekeeping techniques used to increase the efficacy of different veterinary medicines used to treat varroa mites [32]. QC consists of confining the queen in a cage, thus preventing her from laying eggs and artificially inducing a broodless period [33,34]. TC confines the queen to a single frame surrounded by queen excluders. After brood capping, the trapping frame is moved out and destroyed, eliminating a significant number of mites. Once broodless conditions in a honey bee colony are obtained, veterinary medicines are used to kill the varroa mites. By using TC and QC, the acaricidal efficacy of different veterinary medicines, like oxalic acid, can be boosted [33,35,36,37].

This study is the first attempt to assess the viral load of DWV and ABPV before and after using two brood interruption techniques (QC and TC) with an oxalic acid treatment to control varroa mites. We also evaluated the acaricidal efficacy and the effects on the colony strength of the two IPM strategies.

## 2. Materials and Methods

### 2.1. Experimental Sites

Field trials were conducted in two apiaries:Mengeško polje apiary (SI): located in the central Slovenian region (46.146453, 14.575566), characterized by a continental climate. Twenty-two colonies were included in the trials, housed on 10 frames (260 mm × 410 mm). Trials were conducted during the summer season. Day 0 of the protocol was on 12 July 2018, oxalic acid was administered on 6 August 2018, and follow-up treatment began on 21 August 2018.Ciampino apiary (IT): located in the Latium region in central Italy (41.808058, 12.613931), characterized by Mediterranean climate. Here, 26 colonies were housed on 10 Dadant Blatt frame (290 mm × 430 mm) hives. Trials were conducted during the late summer/early autumn seasons. Day 0 of the protocol was on 20 August 2018, oxalic acid was administered on 14 September 2018, and follow-up treatment began on 28 September 2018.

During the experiment, the environmental humidity and temperature were monitored with Ibutton dataloggers (Maxim Integrated, San Jose, CA 95134, USA) placed in the experimental apiaries inside queen cages placed into an empty hive without the bottom tray.

### 2.2. Experimental Setup and Treatments

In each experimental apiary, the colonies were divided into 4 experimental groups:“Queen Caging” (QC) group: queens belonging to this group were caged in VAR-CONTROL^®^ cages (Api-Mo.Bru, Campodoro, Padova, Italy) (Figure 1) from day 0 to day 24. In the SI experimental group, there were 6 colonies, and there were 8 colonies in the IT group.“Trapping comb” (TC) group: queens in this group were caged on a trapping comb (Figure 2) on day 0. On day 20 of the protocol, the queens were transferred into a VAR-CONTROL^®^ cage to evaluate the residual mite fall, and the trapping comb with the capped brood was removed. There were 6 colonies in the SI experimental group and 5 colonies in the IT group.The “Control 1” (CG1) group was established to evaluate the strength of the colonies (SI location: 5 colonies, IT location: 8 colonies).The “Control 2” (CG2) group was established to monitor the natural mite fall (SI location: 5 colonies, IT location: 5 colonies). Queens were caged on day 40 to evaluate the total number of mites killed by follow-up treatment.

### 2.3. Mite Fall

To evaluate the efficacy of the tested protocol, residual mites were treated simultaneously with a single dose of 500 mg of amitraz in strips (Apivar^®^ Veto-pharma, 14 avenue du Québec—ZA de Courtaboeuf, 91140 Villebon-sur-Yvette, France) and a single dose of 800 mg of tau-fluvalinate in strips (Apistan^®^ Vita Europe, Vita House, London Street, Basingstoke RG21 7PG, UK). The follow-up treatment lasted 20 days in the QC and TC groups and 25 days in CG2.

During the whole experiment, the mite fall was monitored every 2–3 days in all colonies by counting mites on bottom boards equipped with sticky sheets. Mites were removed from the sticky sheet after every counting [38].

### 2.4. Colony Strength

The colony strength was assessed three times during the experiment using a grid to count the number of bees on each comb and assessing the area of brood on each frame [39]:-On day 7, to create homogenous groups;-On day 25, to evaluate the impact of the brood interruption techniques;-On day 40, to evaluate the impact of the oxalic acid treatment.

All colonies in groups QC, TC, and CG1 were treated with oxalic acid sucrose solution (Chemicals Laif s.r.l., Vigonza, Italy; dribbling method), according to the manufacturer’s instructions on day 25. The concentration was 4.2% *w*/*v* oxalic acid in 60% *w*/*v* sucrose syrup. The exact experimental protocol is presented in Table 1.

### 2.5. Viral Load

To quantify the viral load, samples of adult worker bees were collected twice, on day 0 and day 25. RNA isolation was performed according to the instructions of a commercial kit (QIAamp^®^ Viral RNA Mini kit-Qiagen (Quiagen SRL, Milan, Italy). Primers and probes are reported in Table 2. Primers targeted a coding sequence of the DWVgp1 gene (NC_004830.2) for DWV detection and a non-coding sequence of the ABPVgp1 gene (NC_002548.1) for ABPV identification. The viral load was assessed using the protocols described in Appendix A.

### 2.6. Acaricide Efficacy

The percentage of acaricide efficacy (AE) in each hive was evaluated using the following formula: AE = VT/(VT + follow-up) × 100, where VT for the QC group represents the total number of mites fallen during the queen caging period and after the subsequent oxalic acid treatment. VT in the TC group represents the total number of mites fallen during the queen caging on the comb, plus the mites found in the trapping frame after its removal, and the mites killed by the oxalic acid treatment. VT + follow-up represents the total number of mites killed by the above-mentioned tested treatment and the follow-up treatments [40].

### 2.7. Statistical Analyses

Kruskal–Wallis and Mann–Whitney tests were used to perform statistical comparisons between different groups with respect to efficacy, strength, and virus load. In the case of comparisons within the same group, the Wilcoxon signed-rank test was used. XLSTAT software (v. 2020.1.3, Addinsoft, Paris, France) was used to perform the statistical analysis.

## 3. Results

### 3.1. Acaricide Efficacy and Cumulative Mite Fall

#### 3.1.1. Slovenian (SI) Apiary

The mean acaricidal efficacy in the TC group was 95.4 ± 1.6%, while the efficacy in the QC group was 81.7 ± 23.4% (Figure 3).

The difference between the TC and QC groups was not statistically significant (*p* = 0.91). The natural mite fall was 5.3 ± 2.2%. The dynamics of the cumulative mite fall can be observed in Figure 4.

#### 3.1.2. Italian (IT) Apiary

The average acaricidal efficacy in the TC group was 96.3 ± 2.5%, which resulted similar to and not statistically different (*p* = 0.876) from the efficacy of the QC group (96.3 ± 3.2%) (Figure 3). The natural mite fall was equal to 53.7 ± 15.8%. The dynamics of the cumulative mite fall can be observed in Figure 4.

### 3.2. Colony Strength

#### 3.2.1. Slovenian (SI) Apiary

The average number of adult bees in CG1 was similar in all three evaluations (mean ± SD) (12,646 ± 5074, 12,464 ± 2712 and 12,092 ± 3105). The differences were not statistically significant (*p* = 0.96). The number of adult bees decreased in the QC and TC groups after the oxalic acid treatment. Statistically significant reductions in the number of adult bees were observed in the QC group between the first and the last checks (V = 22.75; *p* = 0.063) and between the second and the last checks (V = 13.75; *p* = 0.043). In the TC group, the same variations were observed within the same timeframes (V = 22.75; *p* = 0.031).

The average amount of brood in the QC group was 6134 ± 4119 cm^2^, and in TC, it was 10,483 ± 3853 cm^2^. In the CG1 group, the average amount of brood at each time of evaluation was: 9384 ± 3096, 9758 ± 2060, and 6052 ± 2172 cm^2^, respectively, with no statistically significant reductions. All the details are reported in Table A1.

#### 3.2.2. Italian (IT) Apiary

In the QC group, the mean number of bees at each evaluation was 7739 ± 1589, 5892 ± 2849, and 4630 ± 2572, showing a statistically significant decrease after each check (from first to second V = 35.000; *p* = 0.031; from first to last V = 35.000; *p* = 0.016; from second to last; V = 35.000; *p* = 0.031). In the TC group, the average number of bees was 7969 ± 1597, 5429 ± 1438, and 3467 ± 782 at each evaluation time and showed the same decreasing trend (from first to second V = 13.625; *p* = 0.042; from first to last V = 13.750; *p* = 0.063; from second to last V = 13.750; *p* = 0.125). The average number of adult bees in CG1 was always similar. A reduction in bee numbers could be observed in QC and TC on the third strength evaluation, while there was no reduction in the CG1 group (Figure 5).

The average amount of brood in the QC group was 15,060 ± 3964 cm^2^ and in TC 19,985 ± 7743 cm^2^ at the first evaluation time. In the CG1 group, the average amount of brood was 15,677 ± 7443 cm^2^, 19,739 ± 9761 cm^2^, and 20,417 ± 10,819 cm^2^ at the respective strength evaluations. The increase in the brood from the first to the second evaluation was statistically significant (V = 22.750; *p* = 0.046). All the details are reported in Table A1 in Appendix B.

### 3.3. ABPV Loads

#### 3.3.1. Slovenian (SI) Apiary

The mean number of cycles in Real-Time PCR needed to detect ABPV in the QC group before the queen caging was 30.66 ± 0.70 and 34.80 ± 2.58 at the end of the queen caging period. In the TC group, the mean value was 31.68 ± 1.10 before and 33.19 ± 3.63 after the queen was confined in the trapping comb. In the CG1 group, the mean number of cycles to detect ABPV was 29.89 ± 2.50 before and 31.48 ± 7.56 after the applied protocol. In CG2, the mean value was 31.02 ± 1.51 before and 30.40 ± 4.47 after (Figure 6). The ABPV increase in the QC group was statistically significant (V = 22.750; *p* = 0.031).

#### 3.3.2. Italian (IT) Apiary

The mean number of cycles in Real-Time PCR needed to detect ABPV in the QC group before the queen caging was higher (32.10 ± 3.16), and it was lower at the end of the queen caging period (28.50 ± 6.28). In the TC group, we found again a higher number of cycles before queen confinement (31.04 ± 2.39) and a lower number later (29.14 ± 1.09). In the control groups, the mean number of cycles to detect ABPV was 24.21 ± 10.03 before and 26.77 ± 6.56 after the protocol in CG1, and it was 29.52 ± 3.82 before and 26.88 ± 4.39 after in CG2 (Figure 6). The above-mentioned variations of ABPV loads were not statistically significant (TC group *p* = 0.095; QC group *p* = 0.130; CG1 group *p* = 0.805; CG2 group *p* = 0.222).

### 3.4. DWV Loads

#### 3.4.1. Slovenian (SI) Apiary

In the case of the DWV, the mean number of Real-Time PCR cycles in QC was lower before (28.98 ± 8.40) and slightly higher after caging the queen (29.47 ± 2.93). In the TC group, the mean cycles needed to detect the virus before trapping the queen were first higher (32.59 ± 2.15) and then a bit lower (30.63 ± 2.55). In the control groups, we found mean values in CG1 before at 28.46 ± 9.35 and after at 27.54 ± 8.62, and in CG2, there were 27.77 ± 4.80 cycles before and 24.67 ± 9.42 afterward (Figure 7). No statistically significant differences were observed in the DWV loads (TC group *p* = 0.119; QC group *p* = 0.240; CG1 group *p* = 0.690; CG2 group *p* = 1.000).

#### 3.4.2. Italian (IT) Apiary

In the case of the DWV, the mean number of cycles of Real-Time PCR in QC was 16.56 ± 5.18 before and 18.19 ± 7.01 after the queen caging, and in the TC group, 16.14 ± 5.85 cycles were needed before, and 16.85 ± 3.22 were needed after queen confinement to detect the virus. In CG1, the mean value before and after was similar (13.93 ± 7.47 and 13.99 ± 5.32), and in CG2 there were 12.34 ± 2.07 cycles before and 10.39 ± 1.97 after (Figure 7). None of the variations in virus titers was statistically significant (TC group *p* = 0.310; QC group *p* = 0.798; CG1 group *p* = 1.000; CG2 group *p* = 0.222). Summary statistics for a number of Real-Time PCR cycles are reported in Table A2 in Appendix B.

## 4. Discussion

In our study, we tested the acaricide efficacy, the dynamics of varroa mite fall, the colony strength, and the viral load using two integrated varroa management approaches. Brood interruption, by forcing all mites to the phoretic stage, alters the reproductive success of mites [41], and permits increasing the efficacy of the veterinary medicine used. The use of any brood interruption methods also reduces the need for multiple treatments with oxalic acid that are not well tolerated by colonies [42].

As shown in previous studies [33,36,37], the brood interruption techniques are valuable tools not only to increase the acaricidal efficacy of organic acids and essential oils but also to reduce the standard deviation of efficacy among different treated colonies. Our study aimed at further extending the knowledge on the use of brood interruption techniques to control the load of ABPV and DWV, since there are no studies investigating this aspect. The initial infestation levels of varroa were significantly different between the experimental apiaries, higher in the Italian apiary. Mite fall started with the queen confinement and increased after oxalic treatment and reached the maximum level before day 30 according to the protocol in all tested groups. However, during the critical treatment, the mite fall of the control group increased again, which indicates that the efficacy of OA treatment in the presence of brood is lower in comparison to the experimental groups, which is well known [43]. The final varroacide efficacy in our experimental groups was boosted by the absence of a brood induced by the beekeeping techniques. Similar results in the absence of a brood were obtained by other authors [44,45]. The final acaricidal efficacy in the TC in QC groups was very high in both locations. Even if the values were not statistically different from each other, it is interesting to note that the TC technique provided a smaller standard deviation, which may guarantee a more uniform acaricide efficacy among hives. The high acaricidal efficacy in the control group in the Italian apiary could be explained by the higher rate of hygienic behavior in the experimental colonies.

The number of adult bees decreased after the caging period and after the oxalic acid treatment in both the QG and TC experimental groups. Even the control group showed a small decrease in the adult bee population due to the normal seasonal dynamics. The number of bees decreased more evidently in the TC groups. The reduction in the adult bee population is not relevant from a practical point of view, if the colonies are fully developed at the beginning of the caging period. Reassuring results were also obtained by Lodesani and colleagues [37] who found that only two complete brood cycles are needed to fully recover colonies’ strength after brood interruption.

According to Locke and colleagues [29], effective varroa treatments able to indirectly reduce viral load should be carried out a few generations prior to the emergence of winter bees at the end of the summer season. By doing so, the nursing bees of future winter bees will already have a low viral load, thus preventing oral transmission of the virus. Winter bees with a low viral load have a longer lifespan thus increasing the probability for the colony to overwinter successfully [46,47,48]. Our hypothesis was that the viral load after brood interruption would be higher in the QC group than in the control group or TC group due to higher amounts of dispersal mites on adult honey bees. The TC technique leaves brood in the colonies for a longer time than the QC technique offering varroa the possibility to reproduce. Mites are leaving the dispersal phase thus reducing the potential damage related to too high varroa infestation levels on adult bees (including virus transmission from mite to honey bee). Differences in the viral load between the two experimental groups before and after the experiment were observed but were not significant. Only the ABPV increased in the QC group in Slovenia in a statistically significant way. According to Locke and colleagues [29], the DWV titer should decrease immediately after the varroa treatment and gradually start increasing 6 weeks after the treatment. One of the possible reasons why the increase in the viral load in the QC and CG2 groups was not significant might be that the timing between the first and second sampling was too short. Data from Evans and colleagues [49] suggested that a break in brood production through colony swarming significantly reduces mite and DWV levels during the fall (September and October); he observed significant differences in the viral titer by sampling bees after 2–3 months from the break in brood production. Possible reasons for the not significant results could be due to the nature of DWV. DWV infects larvae, pupae, and adults and is found inside eggs. The mortality due to DWV can be seen when a viral load of approximately 10^6^ viral particles/bee is reached. The specific virulence of DWV has repercussions on the number of pupae and adults present in the hives, regardless of the various *Varroa destructor* control techniques; so, all this causes high Ct values in Real-Time PCR.

According to the results observed in the Slovenian apiary, the ABPV titer increased significantly after the QC. It seems that the TC technique may provide more varroa mites the possibility to enter the open cells, leaving the body of the adult bees, and reducing the potential damage linked with too high varroa infestation levels (including virus transmission from mite to bee). In addition, Anderson et al. [50] suggests that the disease first manifests in adults and is then transmitted to larvae.

Differences between the experimental apiaries could be attributed to the different seasonal dynamics of the different honey bee subspecies used. In the Slovenian apiary, *Apis mellifera carnica* was used, and *Apis mellifera ligustica* was used in the Italian apiary. Differences in colony dynamics are reflected also in the mite load and viral load.

Evans [49] suggests that a break in the brood cycle during mid-summer can be effectively used by beekeepers as a non-chemical method of mite control in managed colonies. The brood break should be applied in mid-summer and should end before fall [51] to guarantee colonies enough time to produce winter bees prior to the onset of cold weather. The brood break should not be longer than 21 days in order to let all the brood hatch and to not to weaken the honey bee colony. Our study presented two different brood IPM strategies that could both be very promising not only in boosting the acaricide efficacy of oxalic acid but even in having a reduced viral load in adult bees before wintering. However, further studies should evaluate virus loads after the adoption of the two brood interruption techniques investigated here, prolonging the observation time.

## Figures and Tables

**Figure 1 insects-15-00115-f001:**
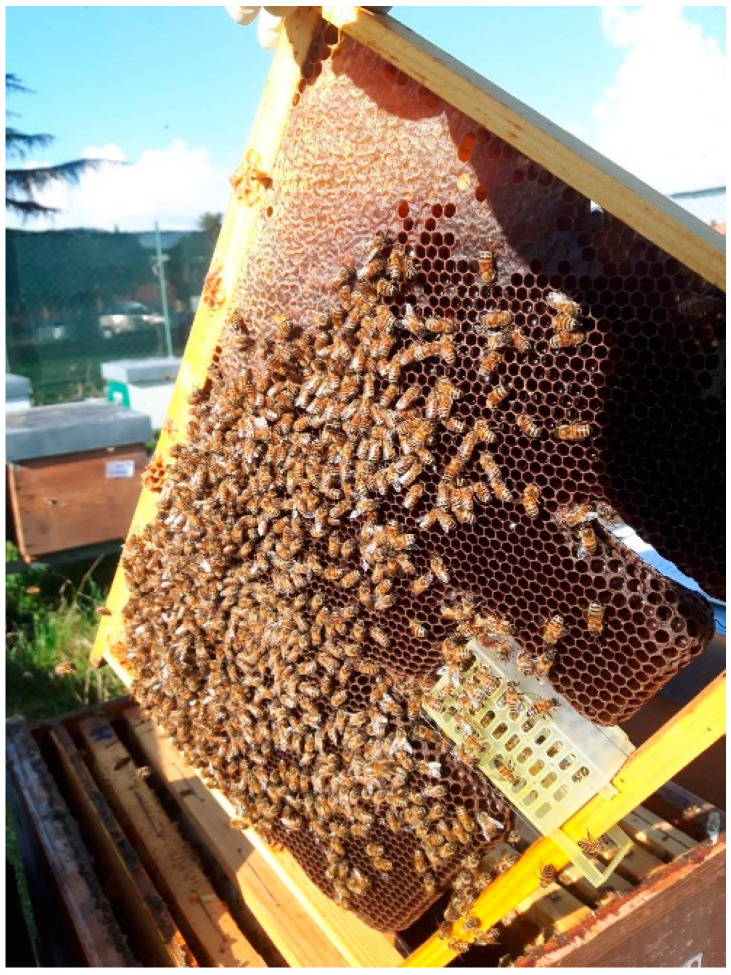
VAR-CONTROL^®^ cage mounted on a frame (Api-Mo.Bru, Campodoro, Padova, Italy).

**Figure 2 insects-15-00115-f002:**
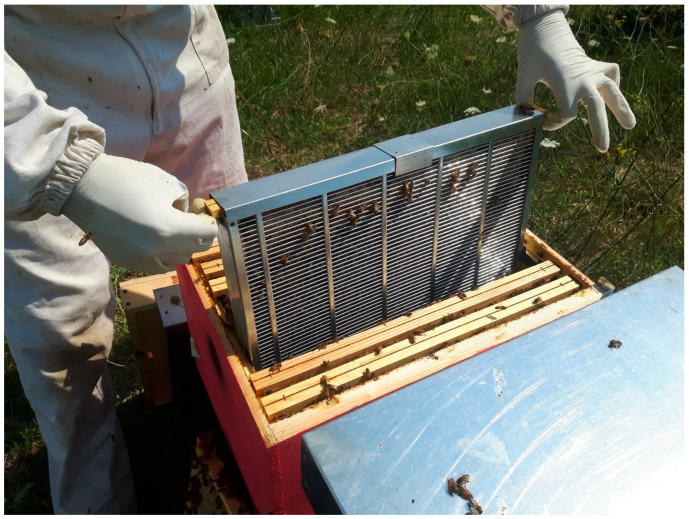
Trapping comb.

**Figure 3 insects-15-00115-f003:**
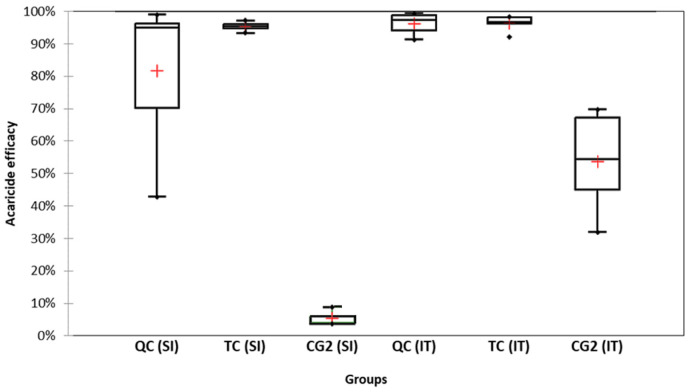
Acaricidal efficacy of applied protocols and natural mortality in control groups. Red crosses are means. QC: queen caging experimental group; TC: trapping comb experimental group; CG2: control group 2. SI stands for Slovenian apiary, and IT stands for Italian apiary.

**Figure 4 insects-15-00115-f004:**
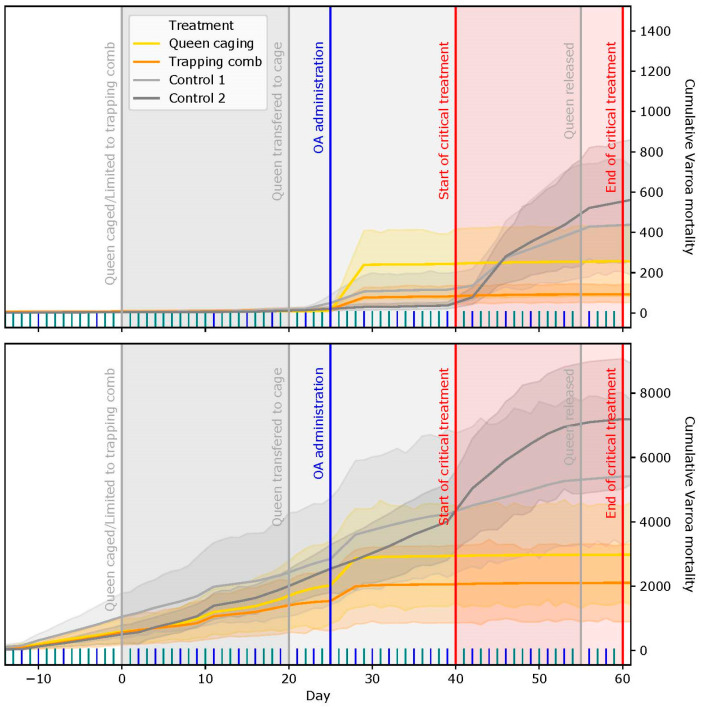
Cumulative mite fall during the experiment. The upper graph represents the data from SI and the lower from IT. The *X* axis represents the days of the protocol, and the *Y* axis represents the number of fallen mites.

**Figure 5 insects-15-00115-f005:**
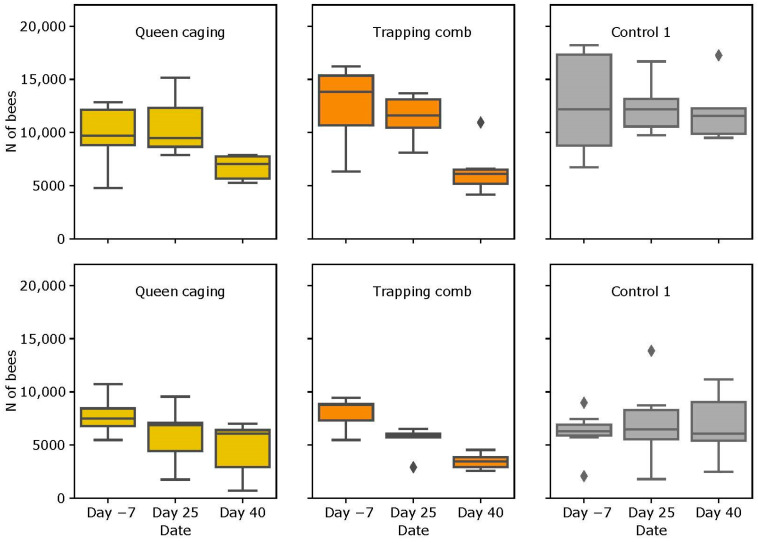
Number of bees per colony for all three groups. The upper line contains the results from the SI apiary, and the lower line contains the results from the IT apiary. Bars represents the standard deviation. Diamonds represent outliers.

**Figure 6 insects-15-00115-f006:**
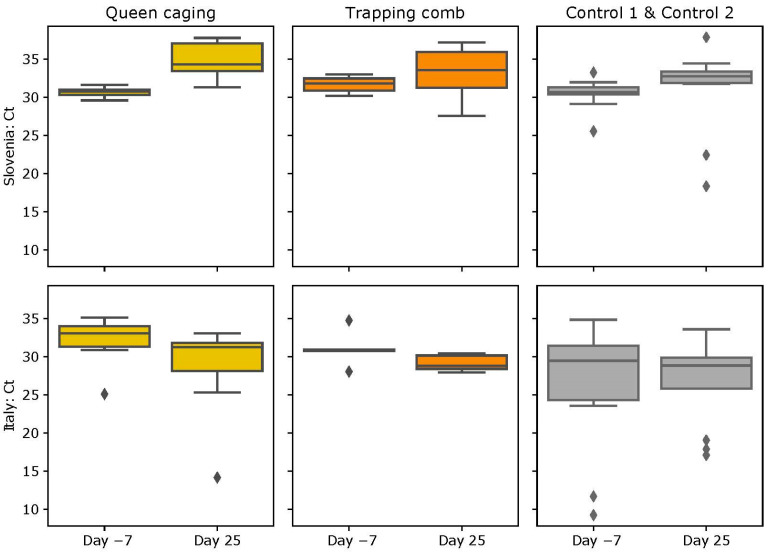
Number of Real-Time PCR cycles needed to detect ABPV. Bars represent the standard deviation. Diamonds represent outliers.

**Figure 7 insects-15-00115-f007:**
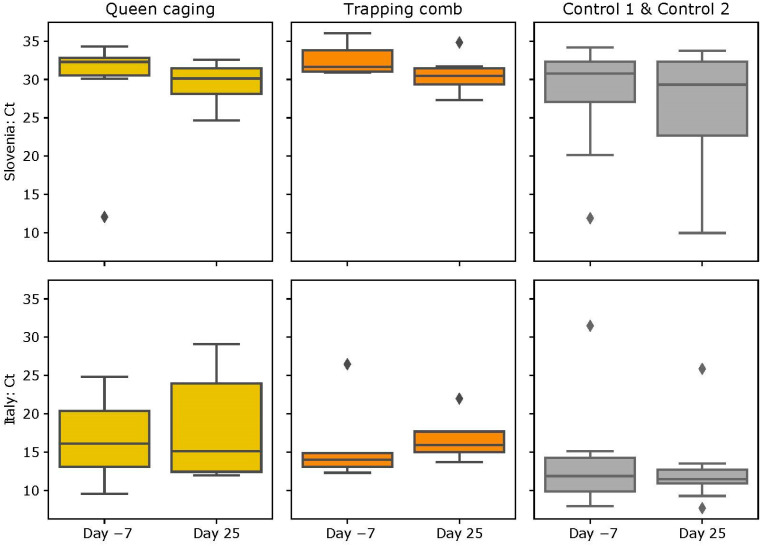
Number of Real-Time PCR cycles needed to detect DWV. Bars represents the standard deviation. Diamonds represent outliers.

**Table 1 insects-15-00115-t001:** Timeline of the experimental protocol applied. Legend: CS: colonies’ strength evaluation; C: caging; SA: sampling for virus loads; TCR: trapping comb removal and transfer of queen into a queen cage; OA: oxalic acid administration; CT: beginning of critical treatment; CE: end of caging; CTE: end of critical treatment; QC: Queen Caging group; TC: Trapping comb group; CG1: Control 1 group; CG2: Control 2 group; End: end of protocol.

		−14	−7	0	20	25	40	55	60	65
GROUPS	QC	Start	CS	C, SA		CS, OA, SA	CT	CE	CTE	End
	TC	Start	CS	C, SA	TCR	CS, OA, SA	CT	CE	CTE	End
	CG1	Start	CS	SA		CS, OA, SA	End			
	CG2	Start	CS	SA		SA	CT, C		CE	CTE, End

**Table 2 insects-15-00115-t002:** Primers and probes used in our experiment.

Primer	Sequence
DWV (Forward)	5′-ATGGGTTTGATTCRATATCTTGGAA-3′
DWV (Reverse)	5′-GATGTTCCRGGTGGCTTTAATGA-3′
DWV Probe	5′-FAM-ACTAGTGCTGGTTTTCCTTTGTC-NFQ-MGB.
ABPV (Forward)	5′-GCCCAGACAAGCGCAGTACT-3′
ABPV (Reverse)	5′-AGCACGGAAAACGCGTCTT-3′
ABPV Probe	5′-FAM-TCCCCGATAGCRACCGA-MGBNFQ-3′

## Data Availability

The data presented in this study are available on request from the corresponding author.

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
