# Peer review of "Integrated Pest Management Strategies to Control Varroa Mites and Their Effect on Viral Loads in Honey Bee Colonies"

_insects, 2024, doi:10.3390/insects15020115_

Round 1
Reviewer 1 Report
Comments and Suggestions for Authors
The ms reported the varroacide efficacy, colonies strength and virus loads after the adoption of two brood interruption techniques on honey bee colonies.
As the health of honeybee is crucial for agricultural production and ecosystem maintenance, the study clearly will contribute to enhancing the honeybee health through proper management of varroa, the most important and damaging honeybee pest.
However at the current status, this manuscript does not fit to the journal standards. Significant major revision is requested for consideration of publishing in the journal.
Simple Summary: The study looks on the effect of the transmission and impacts of the Deformed Wing Virus (DWV) and the Acute Bee Paralysis Virus (ABPV) from Varroa destructor mite on honey bees. ==> I do not see this study on the above mentioned area. Redescribe the background and objectives
20 mite infestation was successfully decreased by both the QC and TC approaches. ==> authors only had checked the mite fall but did not measure any infestation.
In figure 3, TC is more consistent and effective.
In abstract, results are not clearly described. Please do in detail.
39 Both queen caging and trapping comb techniques after an oxalic acid treatment can be considered effective varroa treat- ==> Both queen caging and trapping comb techniques followed by an oxalic acid treatment can be considered effective varroa .....
89 It is the result of a combination of different strategies and measures ==> It is the process of ...
127 and on day 20 of the protocol they were transferred into a VAR-CONTROL® cage,==> Please check if this process is correct, and make the method in detail.
in M&M, some parts are not clear. 1) provide the number of colony replication, 2. provide exact date for OA treatment, follow-up tratment. Also it is not clear if the follow-up treatment is consisted of amitraz and fluvalinate together.
The follow-up treatment lasted 20 days in QC and TC groups and 25 days in CG2. ==> Thus this mean the applied strip was hung for 20 or 25 days and also monitored mite fall through out the period?
It may help to explain if the process of the experiments can be schematically represented.
170 .3. ??
176 The mean acaricidal efficacy in the TC group was 95.4±1.6%, while the efficacy in the QC. This is signficantly high, while how do you understand the high mitefall in the control colonies in IT.
Figure 3. Acarici... ==> Provide the detailed title for the figure so that the statement can be self-explainable for the figure. In all other graphs and table, the same request is being asked.
Figure 4. Wordings are not complete, and visible. Also, provide the unit of the mortality
Table 2. Provide the information on the timing of sampling for colony strength.
I wonder if the information in Table 2 are all from SI?
172 Results are reported separately for each experimental apiary, due to climate differences between Slovenia and Italy.
Be consistent in using SI, IT or apiary naming in the figures and text.
Aos in some graphs and table, IT comes first followed by SI and in others, the other way. Be consistent.
22 since the Varroa mites more successfully left the adult bees and enter open larval cells, which reduced the possibility of harm and virus transmission
141 The follow-up treatment lasted 20 days in QC and TC groups and 25 days in CG2. ==> Looking at the figure 6, the response are in opposite. Not consistent. How do you explain this? With this, extrapolation can be dangerous.
Comments on the Quality of English Language
Need extensive English revision
Author Response
Dear Reviewer,
Thank you for your comments that significantly improved the manuscript. We implemented them in the revised text. Please find below every comment an answer to your comments.
Kind regards
The ms reported the varroacide efficacy, colonies strength and virus loads after the adoption of two brood interruption techniques on honey bee colonies.
As the health of honeybee is crucial for agricultural production and ecosystem maintenance, the study clearly will contribute to enhancing the honeybee health through proper management of varroa, the most important and damaging honeybee pest.
However at the current status, this manuscript does not fit to the journal standards. Significant major revision is requested for consideration of publishing in the journal.
Simple Summary: The study looks on the effect of the transmission and impacts of the Deformed Wing Virus (DWV) and the Acute Bee Paralysis Virus (ABPV) from Varroa destructor mite on honey bees. ==> I do not see this study on the above mentioned area. Redescribe the background and objectives
Thank you for your comment, we re-described background and objectives in the revised manuscript.
20 mite infestation was successfully decreased by both the QC and TC approaches. ==> authors only had checked the mite fall but did not measure any infestation.
The sentence is now correct in the revised manuscript.
In figure 3, TC is more consistent and effective.
The sentence is now correct in the revised manuscript.
In abstract, results are not clearly described. Please do in detail.
Thank you for your comment. We added more detailed results in the abstract section.
39 Both queen caging and trapping comb techniques after an oxalic acid treatment can be considered effective varroa treat- ==> Both queen caging and trapping comb techniques followed by an oxalic acid treatment can be considered effective varroa .....
Thank you for your suggestion. We implemented it in the revised text.
89 It is the result of a combination of different strategies and measures ==> It is the process of ...
Thank you for your suggestion. We implemented it in the revised text.
127 and on day 20 of the protocol they were transferred into a VAR-CONTROL® cage,==> Please check if this process is correct, and make the method in detail.
We explained more in detail the method.
in M&M, some parts are not clear. 1) provide the number of colony replication, 2. provide exact date for OA treatment, follow-up tratment. Also it is not clear if the follow-up treatment is consisted of amitraz and fluvalinate together.
Yes, the follow up treatment consisted of amitraz and fluvalinate together. We explained it better in the revised MS.
The follow-up treatment lasted 20 days in QC and TC groups and 25 days in CG2. ==> Thus this mean the applied strip was hung for 20 or 25 days and also monitored mite fall through out the period?
Yes, mite fall was monitored throughout the whole experiment.
It may help to explain if the process of the experiments can be schematically represented.
Thank you for your suggestion. We included a graphical representation of the experimental protocol.
170 .3. ??
We removed the typo.
176 The mean acaricidal efficacy in the TC group was 95.4±1.6%, while the efficacy in the QC. This is signficantly high, while how do you understand the high mitefall in the control colonies in IT.
Thank you for your comment. We added a more detailed explanation in discussion section.
Figure 3. Acarici... ==> Provide the detailed title for the figure so that the statement can be self-explainable for the figure. In all other graphs and table, the same request is being asked.
We provided the suggested changes in the revised MS.
Figure 4. Wordings are not complete, and visible. Also, provide the unit of the mortality
We provided the suggested changes in the revised MS.
Table 2. Provide the information on the timing of sampling for colony strength.
We provided the suggested changes in section 2.4
I wonder if the information in Table 2 are all from SI?
No, information in Table 2 are from SI and IT. We made it clear more clear in the revised table.
172 Results are reported separately for each experimental apiary, due to climate differences between Slovenia and Italy.
Be consistent in using SI, IT or apiary naming in the figures and text.
Aos in some graphs and table, IT comes first followed by SI and in others, the other way. Be consistent.
We rewrote the sentences using “standard” abbreviations (SI and IT) and we corrected the order in graphs and tables
22 since the Varroa mites more successfully left the adult bees and enter open larval cells, which reduced the possibility of harm and virus transmission
We rewrote it in a clearer way.
141 The follow-up treatment lasted 20 days in QC and TC groups and 25 days in CG2. ==> Looking at the figure 6, the response are in opposite. Not consistent. How do you explain this? With this, extrapolation can be dangerous.
We added a more detailed explanation in discussion.

Reviewer 2 Report
Comments and Suggestions for Authors
Varroa mites may be the greatest problem for beekeeping nowadays. Authors give a brief overview on this problem, mention the major pathogens vectored by the mites and give more detail on the most dangerous viruses which are in the focus of the study. The paper is generally well-written and can be interesting to the readers of the journal, if only authors prove the scientific novelty of this research (see below).
In Introduction, the list of vectored pathogens (Line 52) needs to include microsporidia as a separate group since their inclusion into fungi alongside with other classical fungal taxa such as Aspergillus etc. is a simplification, disseminated by mycologists but not accepted by protistologists. This problem goes beyond the systematics and nomenclature scope as it makes the striking differences in biology and parasitism between classical fungi and fungi-related microsporidia undisclosed. The paragraph in Lines 82-88 is controversary, as it indicates that RNAi has no practical outcome yet but declares it as “the most promising approach”. One has to be more cautious with such conclusions. Moreover, other methods of control are not mentioned, which makes the introduction incomplete. It can also be noted, that authors describe the principles of caging and trapping techniques they use but do not explain how and why the oxalic acid is applied. In Line 78, you mention “the titers of ABPV” citing the paper of 1983, please briefly elaborate on the assessment method so that the readers are aware how it was done then (as compared to now).
In Materials & Methods, the way of oxalic acid application is not clarified, the method for colony strength evaluation is not defined. In Line 144, were the mites removed from sticky sheet each time of counting?
In Results, authors start with a notion that “Results are reported separately for each experimental apiary, due to climate differences between Slovenia and Italy”, but all the relevant Figures and Tables present the respective data both for Italy and Slovenia. Figure 3 contains no explanations for the abbreviations used. In Lines 195-197, it is mentioned that the authors “conducted measurements only once” but provide four consecutive pieces of data (average amount of brood) for the CG1 with no indication what do they correspond to. Moreover, the brood area is 9.384 cm2 in text but 9384 cm2 in Table 2 which is totally confusing. In Figure 7 legend, number of PCR cycles (as in text) is replaced with number of PCR, which is misleading. Finally, there is no comparison between datasets obtained in two different geographic regions.
In Discussion, the second paragraph starts with the phrase “As shown in previous studies [33,36,37], the brood interruption techniques are valuable tools not only to increase acaricidal efficacy of organic acids and essential oils”. It can be therefore concluded that the paper largely reproduced the approaches already utilized and results published. This must be explicitly elucidated in the Introduction to substantiate the scientific novelty of the current manuscript, and novel aspects and conclusions also clarified in Discussion.
Comments on the Quality of English LanguageIn phrases “DWV virus” and “ABPV virus”, the second word is excessive
In Line 138, the indefinite article is missing in “single dose” (see Line 139)
Author Response
Dear Reviewer,
Thank you for your comments that significantly improved the manuscript. We implemented them in the revised text. Please find below every comment an answer to your comments.
Kind regards
Varroa mites may be the greatest problem for beekeeping nowadays. Authors give a brief overview on this problem, mention the major pathogens vectored by the mites and give more detail on the most dangerous viruses which are in the focus of the study. The paper is generally well-written and can be interesting to the readers of the journal, if only authors prove the scientific novelty of this research (see below).
In Introduction, the list of vectored pathogens (Line 52) needs to include microsporidia as a separate group since their inclusion into fungi alongside with other classical fungal taxa such as Aspergillus etc. is a simplification, disseminated by mycologists but not accepted by protistologists. This problem goes beyond the systematics and nomenclature scope as it makes the striking differences in biology and parasitism between classical fungi and fungi-related microsporidia undisclosed.
We corrected the sentence in the revised version of the manuscript.
The paragraph in Lines 82-88 is controversary, as it indicates that RNAi has no practical outcome yet but declares it as “the most promising approach”. One has to be more cautious with such conclusions. Moreover, other methods of control are not mentioned, which makes the introduction incomplete. It can also be noted, that authors describe the principles of caging and trapping techniques they use but do not explain how and why the oxalic acid is applied.
Other methods that are commonly used in practice are mentioned in lines 86,87and 88. In lines 98 and 99 we explain why different veterinary medicines are used.
In Line 78, you mention “the titers of ABPV” citing the paper of 1983, please briefly elaborate on the assessment method so that the readers are aware how it was done then (as compared to now).
In Materials & Methods, the way of oxalic acid application is not clarified, the method for colony strength evaluation is not defined.
Thank you for your comments, we better explained it in the revised text
In Line 144, were the mites removed from sticky sheet each time of counting?
Yes, mites were removed every time when counted. We included this information also in the revised MS.
In Results, authors start with a notion that “Results are reported separately for each experimental apiary, due to climate differences between Slovenia and Italy”, but all the relevant Figures and Tables present the respective data both for Italy and Slovenia.
We rearranged the results section.
Figure 3 contains no explanations for the abbreviations used.
We explained abbreviations on Figure 4 (previously figure 3)
In Lines 195-197, it is mentioned that the authors “conducted measurements only once” but provide four consecutive pieces of data (average amount of brood) for the CG1 with no indication what do they correspond to. Moreover, the brood area is 9.384 cm2 in text but 9384 cm2 in Table 2 which is totally confusing.
Thank you for noticing it. We corrected the concept in the revised version of the MS.
In Figure 7 legend, number of PCR cycles (as in text) is replaced with number of PCR, which is misleading.
We corrected the text in the revised version of the MS.
Finally, there is no comparison between datasets obtained in two different geographic regions.
We added the comparison in discussion section.
In Discussion, the second paragraph starts with the phrase “As shown in previous studies [33,36,37], the brood interruption techniques are valuable tools not only to increase acaricidal efficacy of organic acids and essential oils”. It can be therefore concluded that the paper largely reproduced the approaches already utilized and results published. This must be explicitly elucidated in the Introduction to substantiate the scientific novelty of the current manuscript, and novel aspects and conclusions also clarified in Discussion.
We clarified the novelty of the paper in introduction as well as in discussion section.

Round 2
Reviewer 1 Report
Comments and Suggestions for Authors
The title was changed into IPM to reduce viral load.
I do not fully agree on this, since the study was done explicitly to evaluate the two brood interruption techniques for varroa mite reduction and viral load reduction was acoompanied by the mite management.
This should not be confused by managing viral reduction.
In this sense, please revise the first part of abstract.
The first word in the figure title should be capitalized. (eg. Figure 3. graphical presentation)
Table and figure information are redundant. Please move some data on the table to Appendix or supplementary.
Table 2 ==> Appendix
Table 3 ==> Appendix
Fig. 6. Please detail the y axis title ; Number of bees per colony? What are the bars, SD or SE?
The same to Fig. 7
In discussion,please add the discussion on the duration of brood interruption.
Comments on the Quality of English Language
The title was changed into IPM to reduce viral load.
I do not fully agree on this, since the study was done explicitly to evaluate the two brood interruption techniques for varroa mite reduction and viral load reduction was acoompanied by the mite management.
This should not be confused by managing viral reduction.
In this sense, please revise the first part of abstract.
The first word in the figure title should be capitalized. (eg. Figure 3. graphical presentation)
Table and figure information are redundant. Please move some data on the table to Appendix or supplementary.
Table 2 ==> Appendix
Table 3 ==> Appendix
Fig. 6. Please detail the y axis title ; Number of bees per colony? What are the bars, SD or SE?
The same to Fig. 7
In discussion,please add the discussion on the duration of brood interruption.
Author Response
Dear Reviewer,
thank you for your notes and suggestion. We revised the manuscript accordingly.
Please find below our reply.
Best regards
The title was changed into IPM to reduce viral load.
I do not fully agree on this, since the study was done explicitly to evaluate the two brood interruption techniques for varroa mite reduction and viral load reduction was acoompanied by the mite management.
This should not be confused by managing viral reduction.
In this sense, please revise the first part of abstract.
Thank you for your comment. We modified the title and abstract content
The first word in the figure title should be capitalized. (eg. Figure 3. graphical presentation)
Thank you for the note. We corrected the error.
Table and figure information are redundant. Please move some data on the table to Appendix or supplementary.
Table 2 ==> Appendix
Table 3 ==> Appendix
We moved data to Appendix as suggested
Fig. 6. Please detail the y axis title ; Number of bees per colony? What are the bars, SD or SE?
The same to Fig. 7
We corrected the manuscript according to the comments.
In discussion, please add the discussion on the duration of brood interruption.
We added this info in the revised manuscript.
Reviewer 2 Report
Comments and Suggestions for Authors
The manuscript has been substantially improved but several points still need correction.
The TITLE is misleading, it sounds as a review or at least a study designed specifically to apply IPM strategies to control viral infections with no reference to the primary target of this control – the Varroa mite.
The ABSTRACT exceeds the allowed number of words and contains a thorough presentation of raw data which is appropriate for the Results section. At the abstract level, the readers of the paper should not analyze these data to draw appropriate conclusions, this is the goal of the authors to provide for an informative yet concise summary of obtained results.
The newly added phrase “Our study is the first attempt to control viral infections in honey bees using IPM approach” (L20-21) is a poor explanation of scientific novelty, since the main approach to control viral infections here is the control of the Varroa mite, but the latter has been the target for IPM strategies in a number of works which can be easily discovered using the relevant keywords at scholar.google.com. Thus, authors need to rethink the explanation of scientific novelty of their work in the abstract.
The INTRODUCTION lacks a description of other control methods used for Varroa mite control, other than caging, trapping and oxalic acid
The original comment “In Line 78, you mention “the titers of ABPV” citing the paper of 1983, please briefly elaborate on the assessment method so that the readers are aware how it was done then (as compared to now)” remained unnoticed
In MATERIALS AND METHODS, the concentration of oxalic acid is not indicated
In RESULTS, Figure 3 contains two captions
Author Response
Dear Reviewer,
thank you for your notes and suggestion. We revised the manuscript accordingly.
Please find below our reply.
Best regards
The manuscript has been substantially improved but several points still need correction. The TITLE is misleading, it sounds as a review or at least a study designed specifically to apply IPM strategies to control viral infections with no reference to the primary target of this control – the Varroa mite.
As also suggested by the first reviewer we corrected the title.
The ABSTRACT exceeds the allowed number of words and contains a thorough presentation of raw data which is appropriate for the Results section. At the abstract level, the readers of the paper should not analyze these data to draw appropriate conclusions, this is the goal of the authors to provide for an informative yet concise summary of obtained results.
We modified the abstract according to the suggestions.
The newly added phrase “Our study is the first attempt to control viral infections in honey bees using IPM approach” (L20-21) is a poor explanation of scientific novelty, since the main approach to control viral infections here is the control of the Varroa mite, but the latter has been the target for IPM strategies in a number of works which can be easily discovered using the relevant keywords at scholar.google.com. Thus, authors need to rethink the explanation of scientific novelty of their work in the abstract.
Thank you for your comment. We modified the descriptions of scientific novelty.
The INTRODUCTION lacks a description of other control methods used for Varroa mite control, other than caging, trapping and oxalic acid
We are aware of other methods for varroa control, mainly soft and hard acaricides, and also other methods to control varroa as listed in Rosenkranz et al 2010. However, we think that an overview of those methods would be off topic in our MS.
The original comment “In Line 78, you mention “the titers of ABPV” citing the paper of 1983, please briefly elaborate on the assessment method so that the readers are aware how it was done then (as compared to now)” remained unnoticed
We apologize for not considering your comment. In the past, serological methods were used to detect viruses. Those methods are based on the use of marked antibodies. In case of ABPV, specific antibodies were used to bond virus particles. Since antibodies were marked those markers were later detected. Modern methods are based on detection of nucleic acids of viruses. In case of ABPV, RNA is detected using real time PCR. However, we think that describing method of virus detection is off topic in our introduction.
In MATERIALS AND METHODS, the concentration of oxalic acid is not indicated
Thank you for your comment. We added this info in the revised manuscript.
In RESULTS, Figure 3 contains two captions
Thank you, we corrected the text.